# Towards Easy Vocabulary Drafts with Neologism 2.0

✉ Johannes Lipp[1,3] ⓘ, ✉ Lars Gleim[1] ⓘ, Michael Cochez[2] ⓘ, Iraklis Dimitriadis[3] ⓘ,
Hussain Ali[3] ⓘ, Daniel Hoppe Alvarez[3], Christoph Lange[1,3] ⓘ, and Stefan Decker[1,3] ⓘ

[1] Chair of Information Systems, RWTH Aachen University, Aachen, Germany
[2] Department of Computer Science, Vrije Universiteit Amsterdam, the Netherlands
[3] Fraunhofer Institute for Applied Information Technology FIT, Sankt Augustin, Germany
{last}@dbis.rwth-aachen.de, {first.last}@fit.fraunhofer.de

**Abstract.** Shared vocabularies and ontologies are essential for many applications. Although standards and recommendations already cover many areas, adaptations are usually necessary to represent concrete use-cases properly. Domain experts are unfamiliar with ontology engineering, which creates special requirements for needed tool support. Simple sketch applications are usually too imprecise, while comprehensive ontology editors are often too complicated for non-experts. We present Neologism 2.0 – an open-source tool for quick vocabulary creation through domain experts. Its guided vocabulary creation and its collaborative graph editor enable the quick creation of proper vocabularies, even for non-experts, and dramatically reduces the time and effort to draft vocabularies collaboratively. An RDF export allows quick bootstrapping of any other Semantic Web tool.

**Keywords:** Vocabulary Creation · Ontology Creation · Vocabulary Drafts · Knowledge Graph · Semantic Web.

## 1 Introduction

The benefits of good ontologies and vocabularies are undisputed in the Semantic Web community and the need for semantics is driven by recent trends such as Industry 4.0. General domains received much attention and yielded many recommendations and standards (e.g., SKOS, DCAT, and FOAF). Niche areas, in contrast, often face the problem that no suitable vocabularies exist. At the same time, creating an ontology is a complex, time-consuming task and rarely something domain experts are used to doing – this also holds for lightweight ontologies, which we call "vocabularies". Reducing both effort and ontology expert involvement is crucial, particularly for small-scale application contexts [2]. We identify the key requirements *ease of use* [11], *development speed*, *compatibility with RDF*, *vocabulary reuse*, *ease of publication*, and *collaboration*. This paper presents a remake of *Neologism* [1], which was used to define and publish vocabularies on the US Government Open Data portal[1]. It is available open-source (MIT) at https://github.com/Semantic-Society/Neologism, where we also provide a live demonstration website, a video, and further development towards a fully-featured extension of the original solution. Neologism 2.0 guides users in drafting vocabularies in a graph editor and is, intentionally, not a feature-complete ontology editing tool such as Protégé [13]. It creates vocabulary drafts for bootstrapping other Semantic Web applications.

---

[1]See: https://joinup.ec.europa.eu/node/45149

## 2   Related Work

There are several tools for vocabulary creation, but none that meets all our requirements and fills the gap of creating decent but simple vocabularies easily. The NeOn Toolkit [7] is a Java-based open-source ontology editor that was maintained by the NeOn Foundation until 2011. The AceWiki [10] project enables collaborative ontology management through ACE sentences, a subset of English directly translatable to first-order logic and even more expressive than OWL. SADL [3] enables the definition of semantic models and rules using an English-like DSL, which can be translated to OWL with an Eclipse plugin. Mobi [9] is a free and collaborative knowledge graph platform to publish and discover data models, which is built on RDF and OWL. The fully-featured ontology editor Protégé [13] also provides a web version [16]. The commercial tool TopBraid Composer [15] provides comprehensive support in ontology creation and editing. Another commercial tool called Grafo [4] provides both visual knowledge graph design in a collaborative manner and RDF import/export. ExcelRDF [8] allows RDF vocabulary creation from Excel spreadsheets. CoModIDE [14] combines modular ontology engineering with graphical modeling and is available as a Protégé plugin. Ontology visualization can be done via WebVOWL [12], an interactive web application that supports editing as well [17], or via Protégé plugins that leverage the graph drawing tool Graphviz [5].

A major focus in our related work research is comparing Neologism 2.0 and Neologism 1.0 [1], which both are easy-to-use web-based systems that simplify the process of creating and publishing RDF vocabularies, and use a limited subset of RDFS and OWL. Neologism 2.0 focuses more on a simplified UI to support non-experts by even skipping complex elements like *class disjointness* or *inverse properties*, add autocompleting fields wherever possible (e.g., identifiers). The focus on editing in Neologism 2.0 differs from that on publishing in Neologism 1.0: We provide a graphical editor compared to a viewer. As reliable tools such as Widoco [6] exist that create documentations and visualizations of ontologies, there is no need any more to redundantly implement such features in version 2.0. Besides publishing, Neologism 1.0 used Vapour for validating content negotiation correctness. Consistency checking is out of scope for both versions. Neologism 2.0 intentionally aims at simplifying the creation process for domain users and therefore refrains from overwhelming the user with warnings they might not understand. The quickly created drafts are later repaired and evolved by Semantic Web experts.

In general, the aforementioned tools either lack simplicity or do not meet all our requirements. Fully-featured ontology editors such as Protégé are ideal for ontology experts but overwhelming for users with little to no knowledge about ontologies. As in the comparison with Neologism 1.0, we argue in general that a graphical editor provides strong benefits over text-based editors, particularly for naïve users. Tools like WebVOWL do support graph-based editing, but miss features such as real-time collaboration or publishing ontologies. Additionally, managing and maintaining ontologies in Neologism 2.0 is kept very simple, so that creators can manage access rights to the ontology quite easy and can publish ontologies by themselves. Neologism 2.0 does not focus on only one feature, but combines multiple features, while simplifying them to assist rapid vocabulary creation. We see Neologism 2.0 as proposal to fill the gap between simple sketches and fully-fletched ontology editors, and aim for a strong integration with state-of-the-art tools.

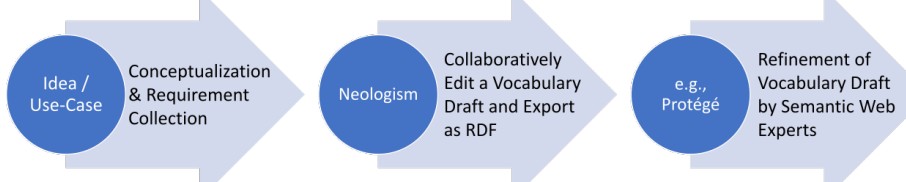

Fig. 1: The vocabulary drafting process embedded into a typical user workflow.

## 3    Embedding Neologism 2.0 in Common Workflows

Most existing solutions have an extensive feature set and are too complex for domain experts who are unfamiliar with the Semantic Web. Neologism 2.0 particularly addresses the requirement of being easy to use in the first iterations of vocabulary creation in specific domains: It provides simple primitives for quick and dirty prototyping of first vocabulary drafts while encouraging best practices and the reuse of existing concepts from configurable sources. Each vocabulary draft is assigned a unique shareable URL that enables easy publishing and collaboration. Figure 1 depicts a suggested workflow that uses Neologism 2.0 for creating early vocabulary drafts and imports its RDF export into Protégé for subsequent refinements by Semantic Web experts, which can finally yield fully-featured ontologies.

## 4    Vocabulary Drafting with Neologism 2.0

Figure 2 depicts the high-level architecture of Neologism 2.0, which consists of the three main components frontend, backend, and recommender. The web frontend is written in TypeScript and uses Angular 9.0, and offers authentication and authorization. A simple dashboard allows managing, creating, editing, sharing, searching, and exporting vocabularies. Data persistence is managed by the Meteor backend, which serves REST endpoints for edit operations on classes and properties. As a major feature, Neologism 2.0 immediately synchronizes changes among all clients and the server to enable real-time collaboration, which avoids confusion among users caused by asynchronous edits.

Users create vocabularies in Neologism 2.0's graphical editor; see the screenshots in Figure 3. Only name and description are needed to create new classes, while the URI is autocompleted from a generated ontology base URI and the class name.

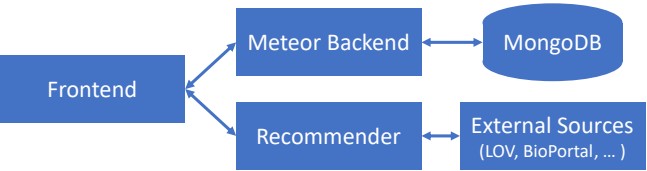

Fig. 2: Neologism 2.0's frontend communicates with a backend to persist information, and might interact with a recommender to improve the quality of drafted vocabularies.

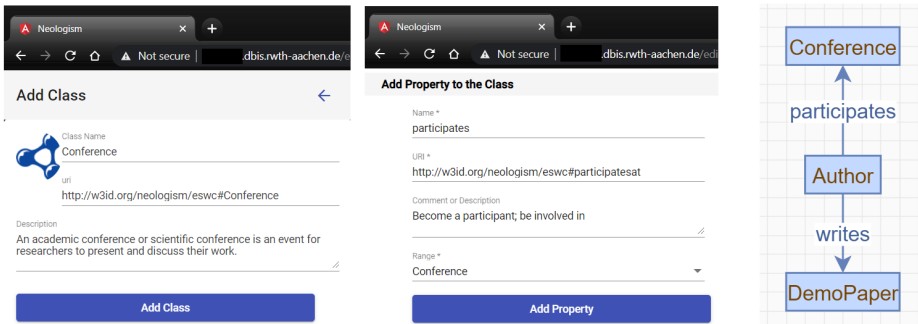

Fig. 3: Users add new classes via name and description (left), or add properties by choosing a range and entering name and description to a popup (center). The graph editor (right) is interactive. All URIs are generated automatically to ease access.

The same holds for properties, which additionally require a range definition. User input from the web frontend later adds to the ontology export as follows. We use `rdf:type` to define `rdfs:Class`, `owl:Class`, and `owl:ObjectProperty`. The names and descriptions of classes and properties in the graph editor are serialized as `rdfs:label`/`comment`, respectively. We finally use `rdfs:domain` and `rdfs:range` to express each property's nodes in the generated export RDFS/OWL file in turtle format.

The graph editor depicts the current state and allows interactions on both classes and properties, including editing and deletion. It supports real-time collaboration by continuously synchronizing its state. The work-in-progress recommender suggests existing terms from configurable sources, depending on the deployment. It can also act as a proxy to recommendation systems such as LOV or BioPortal.

**Limitations.** Motivated by a focus on simplicity for quick ontology drafting, Neologism 2.0 intentionally omits support for more complex OWL semantics, such as cardinality restrictions etc., thus restricting its applicability as a general purpose ontology editor compared to related work. However, this restriction by design is in line with the drafting process depicted in Fig. 1. Furthermore, Neologism 2.0 does not currently support importing existing RDF vocabularies for further drafting.

## 5    Future Work

The open-source vocabulary prototyping tool Neologism 2.0 fits well in Semantic Web workflows where domain experts create first drafts, which are later revised in more comprehensive tools by Semantic Web experts. The current state covers all key requirements except for the following *ease of publication* and *vocabulary reuse*. Ongoing research will tackle the former by adding the features of the initial Neologism tool, and the latter by enhancing the recommender's consideration of the context of ontologies to improve the quality of designs and thus facilitate subsequent ontology development. Finally, a user study in different Semantic Web projects from the production and mobility sectors will evaluate both usability and integration in real-world use-cases. Our open-source GitHub repository allows collaboration and discussions with the community to close the

gap between simple sketches and comprehensive ontology editors and finally improve collaboration between experts from the domain and the Semantic Web.

**Acknowledgment: Funded by the Deutsche Forschungsgemeinschaft (DFG, German Research Foundation) under Germany's Excellence Strategy – EXC – 2023 Internet of Production – 390621612.**

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
