# OpenReview forum: "Towards Easy Vocabulary Drafts with Neologism 2.0"
_eswc-conferences.org/ESWC/2021/Conference/Poster_and_Demo_Track — ESWC2021 P&D_

### Official Review · AnonReviewer1 · 2021-04-15
**a possibly interesting ontology editor**

**Rating:** 6
**Confidence:** 4

**Review:**

This paper presents an open-source tool to develop vocabularies, possibly collaboratively, for non SW experts.
The paper is well written and interesting for the ESWC community. The source code and documentation is freely available on github.
The state of the art is well summarized. The positioning of the presented tool with respect to the state of the art is not precise enough. The positioning with respect to its former version is unclear too.
A user study is necessary, but it could come in an upcoming longer paper.
The model of the created vocabularies is unclear. It is mentioned several times in the paper that the tool allows an RDF export. It should better be clarified that it enables an OWL export (if it is the case).



**Anonymity:**

Yes, I would like my review to remain anonymous.

---

### Official Review · AnonReviewer4 · 2021-04-15
**Good match**

**Rating:** 8
**Confidence:** 3

**Review:**

This contribution presents an open-source tool for vocabulary creation, which allows non-experts to create vocabularies.
The article starts with an excellent introduction, which points at difficulties non-experts face when designing lightweight ontologies.
The software described in the paper is freely available and has a track record of being used in a previous important application. The authors go beyond providing a comprehensive tool by suggesting a workflow for incorporating it into common workflows.  The paper provides also technical information on the implementation. As for possible improvements, the authors could consider providing a section covering limitations and comparison with other tools, the delta against the previous version of Neo could be more precisely stated.
 Since this submission is supplemented by a video demo, it is in my opinion excellent match for the conference.

**Anonymity:**

Yes, I would like my review to remain anonymous.

---

### Official Review · AnonReviewer2 · 2021-04-16
**It is not enough clear the novelty of the proposed system while I can understand the authors' motivations.**

**Rating:** 5
**Confidence:** 4

**Review:**

This paper presents a collaborative vocabulary creation tool called Neologism 2.0.
The authors’ motivation is providing a tool to create  draft vocabulary not for ontology engineers but  domain experts.
Therefore, the tool is designed under the idea that most of existing ontology editing tools are too complex for domain experts.
I agree that this idea is very important and the basic functions of the proposed system are well-designed.
However, it is not enough clear which functions are novel in compare to existing tools.
The authors list many ontology editing tools in related works section, but they are many tool for creating not full-ontologies but draft vocabularies. In my understanding, many of them provide almost same functions with  the proposed tool.
In particular,  what is the new feature of the tool from Neologism 1.0?





**Anonymity:**

Yes, I would like my review to remain anonymous.

---

### Decision · Program_Chairs · 2021-04-19

Accept